# Firing rate-dependent phase responses of Purkinje cells support transient oscillations

**Yunliang Zang, Sungho Hong, Erik De Schutter\***

Computational Neuroscience Unit, Okinawa Institute of Science and Technology Graduate University, Okinawa, Japan

**Abstract** Both spike rate and timing can transmit information in the brain. Phase response curves (PRCs) quantify how a neuron transforms input to output by spike timing. PRCs exhibit strong firing-rate adaptation, but its mechanism and relevance for network output are poorly understood. Using our Purkinje cell (PC) model, we demonstrate that the rate adaptation is caused by rate-dependent subthreshold membrane potentials efficiently regulating the activation of $Na^+$ channels. Then, we use a realistic PC network model to examine how rate-dependent responses synchronize spikes in the scenario of reciprocal inhibition-caused high-frequency oscillations. The changes in PRC cause oscillations and spike correlations only at high firing rates. The causal role of the PRC is confirmed using a simpler coupled oscillator network model. This mechanism enables transient oscillations between fast-spiking neurons that thereby form PC assemblies. Our work demonstrates that rate adaptation of PRCs can spatio-temporally organize the PC input to cerebellar nuclei.

**\*For correspondence:**
erik@oist.jp

**Competing interests:** The authors declare that no competing interests exist.

## Introduction

The propensity of neurons to fire synchronously depends on the interaction between cellular and network properties (*Ermentrout et al., 2001*). The contribution of cellular properties can be measured with a phase response curve (PRC). The PRC quantifies how a weak stimulus exerted at different phases during the interspike interval (ISI) can shift subsequent spike timing in repetitively firing neurons (*Ermentrout et al., 2001*; *Ermentrout et al., 2012*; *Gutkin et al., 2005*) and thereby predicts how well-timed synaptic input can modify spike timing. Consequently, the PRC determines the potential of network synchronization (*Ermentrout et al., 2001*; *Ermentrout et al., 2008*; *Gutkin et al., 2005*; *Smeal et al., 2010*). However, the PRC is not static and shows significant adaptation to firing rates. In cerebellar Purkinje cells (PCs), their phase responses to weak stimuli at low firing rates are small and surprisingly flat. With increased rates, responses in later phases become phase-dependent, with earlier onset-phases and gradually increasing peak amplitudes. This PRC property has never been theoretically replicated or explained (*Couto et al., 2015*; *Phoka et al., 2010*), nor has its effect on synchronizing spike outputs been explored.

On the circuit level, high-frequency oscillations caused by reciprocal inhibition have been observed in many regions of the brain, including the cortex, cerebellum and hippocampus (*Bartos et al., 2002*; *Buzsáki and Draguhn, 2004*; *Cheron et al., 2004*; *de Solages et al., 2008*). The functional importance of oscillations in information transmission is largely determined by their spatio-temporal scale, which for hard-wired inhibitory connections, is generally assumed to be driven by external input. It is interesting to explore whether firing rate-dependent PRCs can contribute to dynamic control of the spatial range of oscillations based on firing rate changes, because this would have significant downstream effects (*Person and Raman, 2012*).

To examine the mechanism of rate-dependent PRCs, we use our physiologically detailed PC model (*Zang et al., 2018*) and a simple pyramidal neuron model to explore the rate adaptation of PRCs. By analyzing simulation data and in vitro experimental data (*Rancz and Häusser, 2010*), we show that rate-dependent subthreshold membrane potentials can modulate the activation of $Na^+$ channels to shape neuronal PRC profiles. We also build a PC network model connected by inhibitory axon collaterals to simulate high-frequency oscillations (*de Solages et al., 2008*; *Witter et al., 2016*). Rate adaptation of PRCs increases the power of oscillations at higher firing rates, firing irregularity and network connectivity also regulate the oscillation level. The causal role of the PRC is confirmed using a simpler coupled oscillator network model. The combination of these factors enables PC spikes uncorrelated at low basal rates to become transiently correlated in transient assemblies of PCs at high firing rates.

## Results

### PRC exhibits rate adaptation in PCs

PRCs were obtained by repeatedly exerting a weak stimulus at different phases of the ISI. The resulting change in ISI relative to original ISI corresponds to the PRC value at that phase (*Figure 1A*). All previous abstract and detailed PC models failed to replicate the experimentally observed rate adaptation of PRCs (*Akemann and Knöpfel, 2006*; *Couto et al., 2015*; *De Schutter and Bower, 1994*; *Khaliq et al., 2003*; *Phoka et al., 2010*). Our recent PC model was well constrained against a wide range of experimental data (*Zang et al., 2018*). Here, we explored whether this model can capture the rate adaptation of PRCs under similar conditions. When the PC model fires at 12 Hz, responses (phase advances) to weak stimuli are small and nearly flat for the whole ISI (*Figure 1B,D*). Only at a very narrow late phase do the responses become phase-dependent and slightly increased. With increased rates, the responses remain small and flat during early phases. However, later phase-dependent peaks gradually become larger (*Figure 1C*), with onset shifted to earlier phases (*Figure 1D*). It should be noted that the increased late-peak amplitude may be affected by how the PRC is computed (*Equation 1*): it is normalized by the ISI, causing the peak amplitude to increase for higher firing rates (smaller ISIs).

In agreement with experiments under the same stimulus conditions (*Phoka et al., 2010*), the peak of PRCs finally became saturated at ~0.06 at high rates. The relationship between normalized PRC peaks and rates can be fitted by the Boltzman function and matches experimental data (*Figure 1C*, fitted with $1/(1 + e^{-(rate-a)/b})$, a = 49.1, b = 26.4 in the model versus a = 44.1 and b = 20.5 in experiments *Couto et al., 2015*). PRCs in our model show similar rate adaption with inhibitory stimuli (phase delay, *Figure 1—figure supplement 1A*). This form of rate adaptive PRCs requires the presence of a dendrite in the PC model (*Figure 1—figure supplement 2*), but the dendrite can be passive (*Figure 1—figure supplement 1B*). We also tested the effect of increasing stimulus amplitude on PRC adaptation. Increasing stimulus amplitude consistently shifts onset-phases of phase-dependent peaks to the left and increases their amplitudes (*Figure 1—figure supplement 1C*).

To unveil the biophysical principles governing rate adaptive PRC profiles, we need to answer two questions: why are responses nearly flat in early phases and why do responses become phase-dependent during later phases?

### The biophysical mechanism of rate adaptation of PRCs in PCs

We examined how spike properties vary with firing rates and find that the facilitation of $Na^+$ currents relative to $K^+$ currents, due to elevated subthreshold membrane potentials at high rates, underlies the rate adaptation of PRCs. After each spike, there is a pronounced after-hyperpolarization (AHP) caused by the large conductance $Ca^{2+}$-activated $K^+$ current, and subsequently the membrane potential gradually depolarizes due to intrinsic $Na^+$ currents and dendritic axial current (*Zang et al., 2018*). As confirmed by re-analyzing in vitro somatic membrane potential recordings (shared by Ede Rancz and Michael Häusser *Rancz and Häusser, 2010*), subthreshold membrane potential levels are significantly elevated at high firing rates, but spike thresholds rise only slightly with rates (*Figure 2A*). This means that the ISI phase where $Na^+$ activation threshold (~ −55 mV for 0.5% activation in PCs *Khaliq et al., 2003*; *Zang et al., 2018*) is crossed shifts to earlier phases with

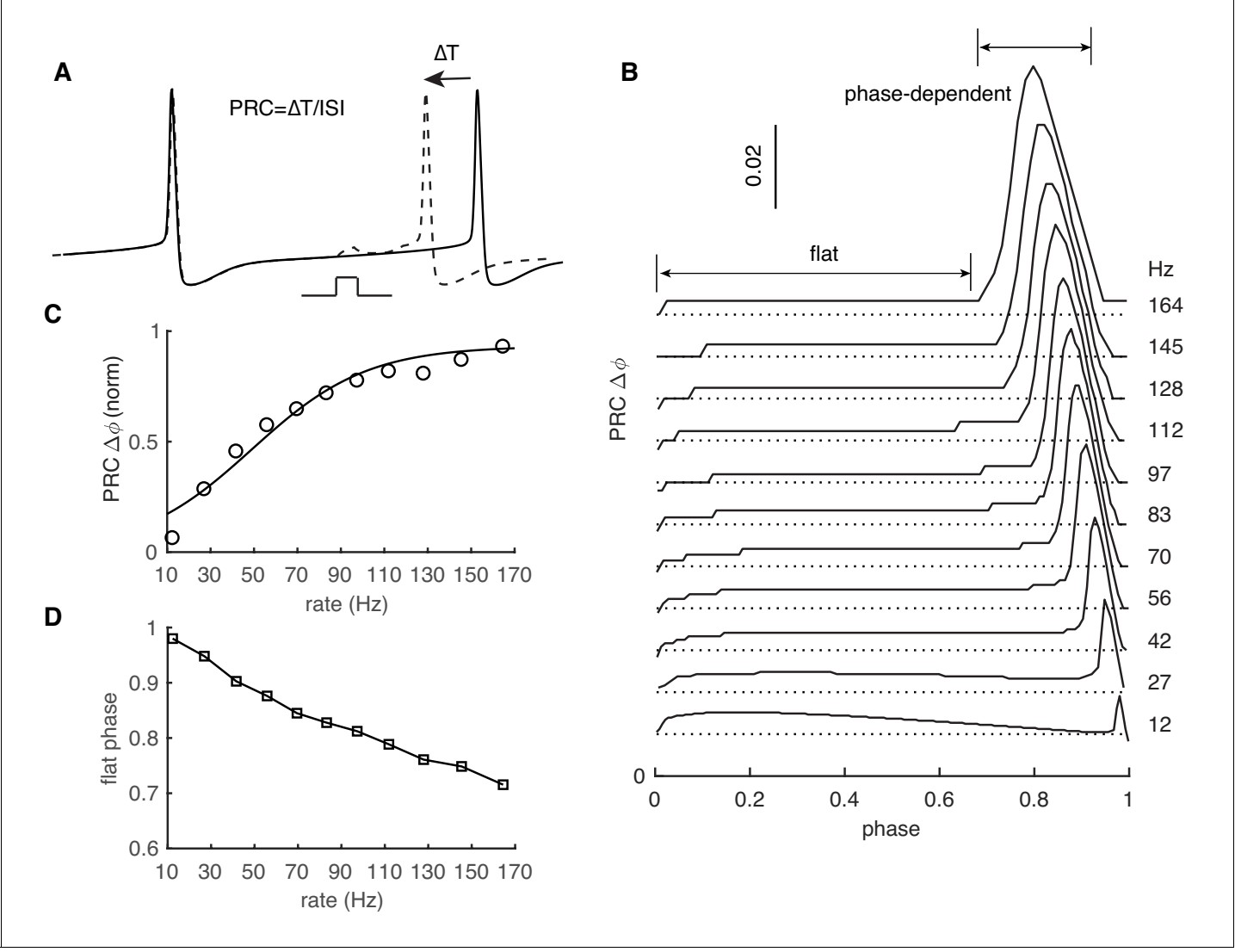

**Figure 1.** PRC exhibits strong rate adaptation in PC model. (A) Schematic representation of the definition and computation of PRCs. The current pulse has a duration of 0.5 ms and an amplitude of 50 pA. Different spike rates were achieved by somatic current injection (*Couto et al., 2015*; *Phoka et al., 2010*). (B) The rate adaptation of the flat part and the phase-dependent PRC peak. (C) PRC peak amplitudes at different firing rates fitted by the Boltzmann function. (D) Duration of the flat phase at different firing rates.

The online version of this article includes the following figure supplement(s) for figure 1:

**Figure supplement 1 .** Rate-dependent PRCs.

**Figure supplement 2 .** Rate-dependent PRCs are influenced by the dendrite.

increasing rates. Consequently, larger phase ranges of membrane potentials are above the threshold at high rates (*Figure 2B*).

During early phases of all firing rates, membrane potentials are distant from the $Na^+$ activation threshold of the $Na^+$ channels (*Figure 2A,B*). The depolarizations to weak stimuli fail to activate sufficient transient and resurgent $Na^+$ channels to speed up voltage trajectories (*Figure 2C*). Consequently, phase advances in early phases are small and flat. At later phases, membrane potentials gradually approach and surpass the $Na^+$ activation threshold. Stimulus-evoked depolarizations activate more $Na^+$ channels to speed up trajectories in return. Therefore, phase advances become large and phase- (actually voltage-) dependent. Because high-rate-corresponding elevated membrane potentials have larger slopes at the foot of the $Na^+$ activation curve, the same $\Delta V$ activates more $Na^+$ channels and, in addition to the normalization, contributes to larger PRC peaks at high rates

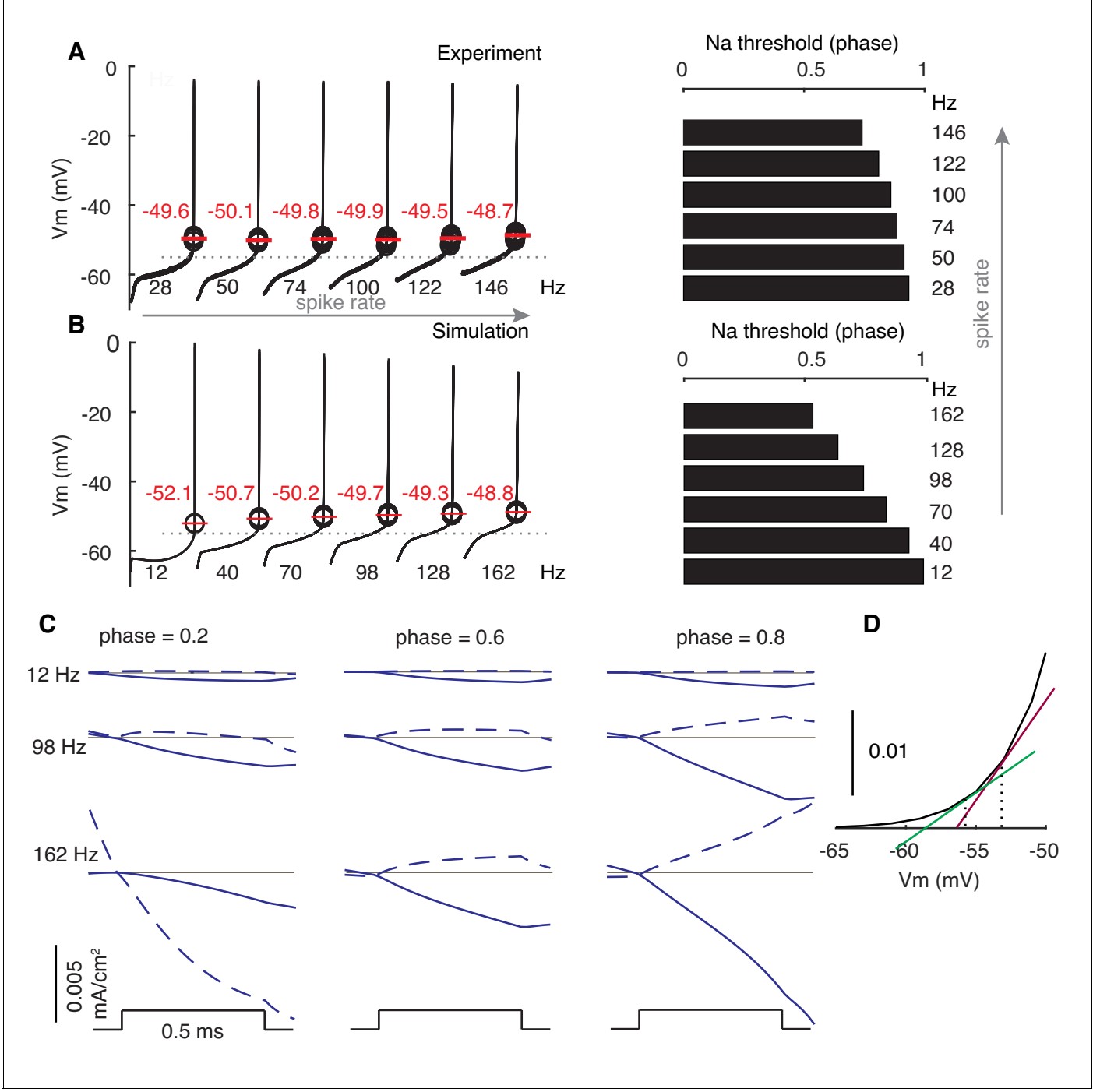

**Figure 2.** Modulated subthreshold membrane potentials account for the rate adaptation of PRCs. (A and B) Experimental and simulated voltage trajectories in PCs at different rates. All voltage trajectories are shown from trough to peak within normalized ISIs. The model used (*Zang et al., 2018*) was not fitted to this specific experimental data. Spike thresholds at different rates are labeled in plots. The $Na^+$-activation threshold is defined as −55 mV (stippled line). Right plots show phase dependence of $Na^+$-activation threshold on firing rates. (C) Stimulus-triggered variations of inward ionic currents (solid) and outward ionic currents (dashed) at different phases and rates. Ionic currents are shifted to 0 (grey line) at the onset of stimulus to compare their relative changes. At phase = 0.2, the outward current is still decreasing due to the inactivation of the large conductance $Ca^{2+}$-activated $K^+$ current at 162 Hz. (D) Larger slopes of the $Na^+$ activation curve at high membrane potentials.

The online version of this article includes the following figure supplement(s) for figure 2:

**Figure supplement 1 .** Effect of Subthreshold Membrane Potentials on Shaping PRCs.

(*Figure 2C,D*). Under all conditions (except phase = 0.2, 162 Hz), stimulus-evoked depolarizations also increase outward currents, but this increase is smaller than that of inward currents (mainly $Na^+$) due to the high activation threshold of $K^+$ currents (mainly Kv3) in PCs (*Martina et al., 2003*; *Zang et al., 2018*). As the stimulus becomes stronger, it triggers larger depolarizations and the required pre-stimulus membrane potential (phase) to reach $Na^+$ activation threshold is lowered. Thus, increasing the stimulus amplitude not only increases PRC peaks, but also shifts the onset-phases of phase-dependent responses to the left (*Figure 1—figure supplement 1C*). In the absence of a dendrite (*Figure 1—figure supplement 2*), the larger amplitude spike is followed by a stronger afterhyperpolarization (*Zang et al., 2018*) that deactivates $K^+$ currents allowing for an earlier depolarization in the ISI, resulting in a completely different PRC.

We further confirmed that the critical role of subthreshold membrane potentials in shaping PRC profiles is not specific to the PC by manipulating PRCs in a modified Traub model (*Ermentrout et al., 2001*; *Figure 2—figure supplement 1* and accompanying text).

## Rate-dependent high-frequency oscillations

The potential effect of firing rate-caused variations of cellular response properties on population synchrony has been largely ignored in previous studies (*Bartos et al., 2002*; *Brunel and Hakim, 1999*; *de Solages et al., 2008*; *Heck et al., 2007*; *Shin and De Schutter, 2006*). Here, we examine whether spike rate correlates with synchrony in the presence of high-frequency oscillations that have been observed in the adult cerebellar cortex (*Cheron et al., 2004*; *de Solages et al., 2008*). Although axon collateral contacts between PCs were originally described to exist only in juvenile mice (*Watt et al., 2009*), recent work demonstrated their existence also in adult mice (*Witter et al., 2016*). We built a biophysically realistic network model composed of 100 PCs with passive dendrites distributed on the parasagittal plane (*Witter et al., 2016*). Each PC connects to the somas of its five nearest neighboring PCs through inhibitory axon collaterals on each side based on experimental data (*Bishop and O'Donoghue, 1986*; *de Solages et al., 2008*; *Watt et al., 2009*; *Witter et al., 2016*). Rates of each PC are independently driven by parallel fiber synapses, stellate cell synapses, and basket cell synapses (*Figure 3A*). More details are in Materials and methods.

When the average cellular rate is 116 Hz, PCs in the network tend to fire within interspaced clusters with time intervals of ~6 ms (*Figure 3B*). However, individual PCs do not fire within every cluster. Therefore, spikes in the network show intermittent pairwise synchrony on the population level rather than spike-to-spike synchrony (*Figure 3B*). Each peak in *Figure 3C* corresponds to a 'cluster'. Based on the power spectrum, the network oscillates at a frequency of ~175 Hz (inverse of the cluster interval, ~6 ms), which is independent of cellular firing rates (116 Hz in red and 70 Hz in blue, *Figure 3D*), because oscillation frequency is mainly determined by synaptic properties (*Brunel and Hakim, 1999*; *Brunel and Wang, 2003*; *de Solages et al., 2008*; *Maex and De Schutter, 2003*). When cellular firing rates increase from 70 Hz to 116 Hz, the power of high-frequency oscillations significantly increases and the peak becomes sharper. When individual PCs fire at low rates (10 Hz), the network fails to generate high-frequency oscillations and each PC fires independently, as evidenced by the flat power spectrum (*Figure 3D*). High-frequency oscillations and their firing rate-dependent changes are also reflected in the average normalized cross-correlograms (CCGs) between PC pairs (*Figure 3E*). When PCs fire at 70 Hz and 116 Hz, in addition to positive central peaks, two significant side peaks can be observed in the CCGs, suggesting correlated spikes with 0 ms-time lag and ~6 ms-time lag. Amplitudes of the peaks increase with cellular firing rates and disappear when they are low (10 Hz).

In *Figure 3*, the variation of cellular rates was driven by synaptic input to demonstrate the rate adaptation of high-frequency oscillations. However, it is difficult to differentiate the relative contribution of PRC shapes and firing irregularity (measured by the CV of ISIs) since they covary with firing rates (*Figure 3D*). Therefore, cellular rates were systematically varied by dynamic current injections, which were approximated by the Ornstein–Uhlenbeck (OU) process (*Destexhe et al., 2001*). This simulation protocol also causes the formation of high-frequency oscillations (*Figure 4—figure supplement 1*). When PCs fire with low to moderate CV of ISIs, they show loose spike-to-spike synchrony at high rates, and the power peak increases with cellular firing rates. High-frequency oscillations were never observed for low cellular firing rates (*Figure 4A*, *Figure 4—figure supplement 1*). With high CV of ISIs, spikes are jittered and the loose spike synchrony is disrupted (*Figure 4B*). Oscillation changes due to firing properties are also reflected in average normalized

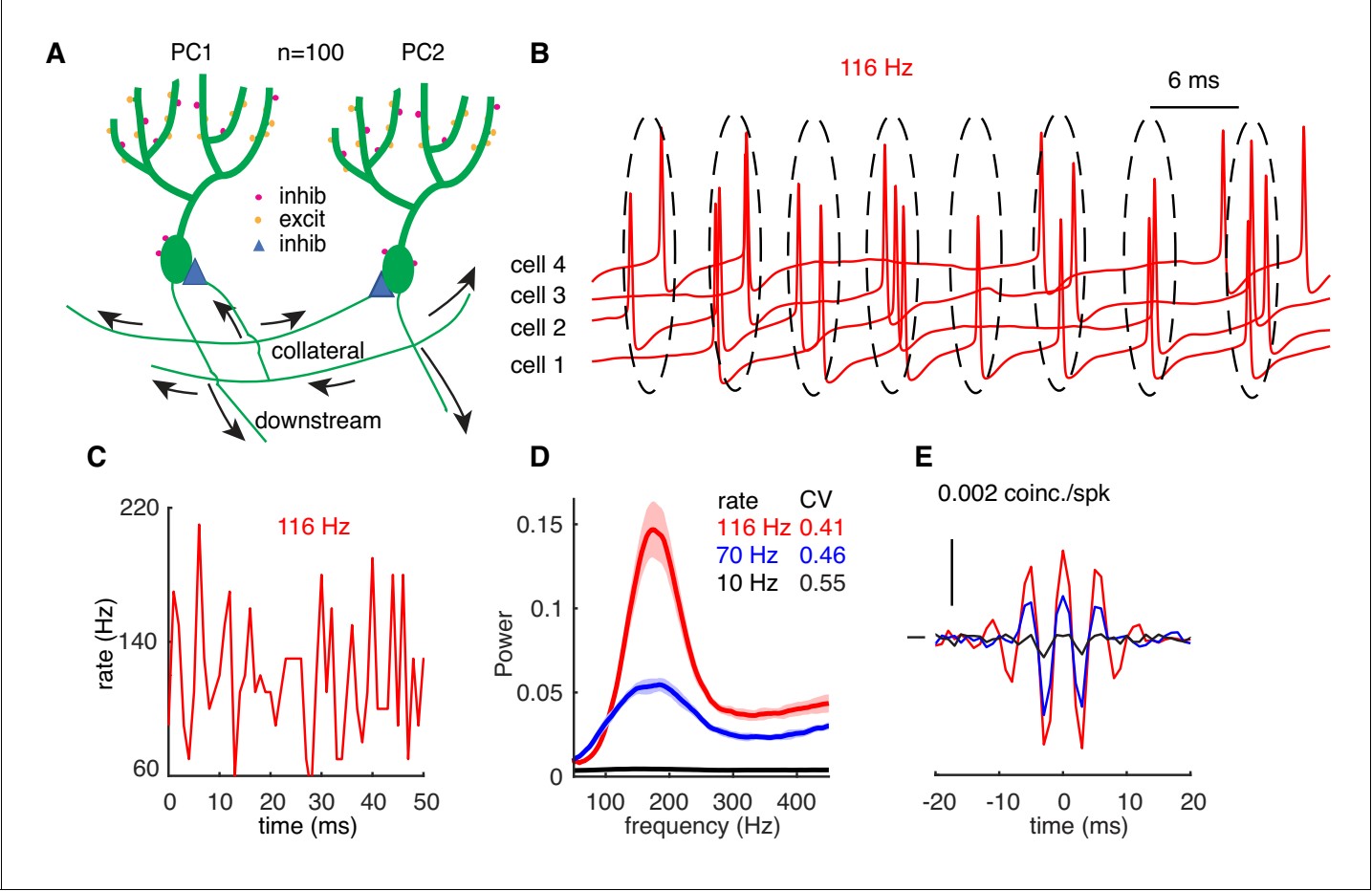

**Figure 3.** High-frequency oscillations show adaptation to cellular firing rates. (**A**) Schematic representation of the network configuration. (**B**) Example of sampled PC voltage trajectories in the network. (**C**) Example of population rates in the network (time bin 1 ms). (**D**) The power spectrum of population rates of the network at different cellular rates and firing irregularity (CV of ISIs). (**E**) Averaged normalized CCGs at different cellular rates.

CCGs. Both central and side peaks increase with the cellular firing rate and decrease with the spiking irregularity. Our results show that small spiking irregularity supports high-frequency oscillations.

At the circuit level, the strength of inhibitory synapses and connection radius are difficult to determine accurately, but their values are critical for the function of axon collaterals. Within the ranges of experimentally reported synaptic conductance and connection radius (*de Solages et al., 2008*; *Orduz and Llano, 2007*; *Watt et al., 2009*; *Witter et al., 2016*), the network generates robust high-frequency oscillations (*Figure 4C,D*). In addition, we find that increasing the conductance of inhibitory synapses or their connection radius increases the power of high-frequency oscillations and make the power spectrum sharper. The increased oscillation power due to connectivity properties is also captured by the larger peaks in CCGs.

Together, our simulation data suggest that the correlation between PC spikes is strong under conditions of low to moderate spiking irregularity, high cellular firing rate, high synaptic conductance, and large connection radius.

## High-frequency oscillations are caused by rate-dependent PRCs

Because both oscillation power and PRC are firing rate dependent, a causal relationship is possible. This is supported by the effect of PRC size on oscillations: decreasing its size leads to weaker oscillations and can even cause weaker oscillations at higher spike rates (*Figure 4—figure supplement 2*). However, it is impossible to manipulate PRC shapes in the complex PC model without greatly affecting other cell and network properties. Therefore, we investigated the effect of rate-dependent PRC shapes in a network of simple coupled oscillators (*Kuramoto, 1984*; *Smeal et al., 2010*), where the

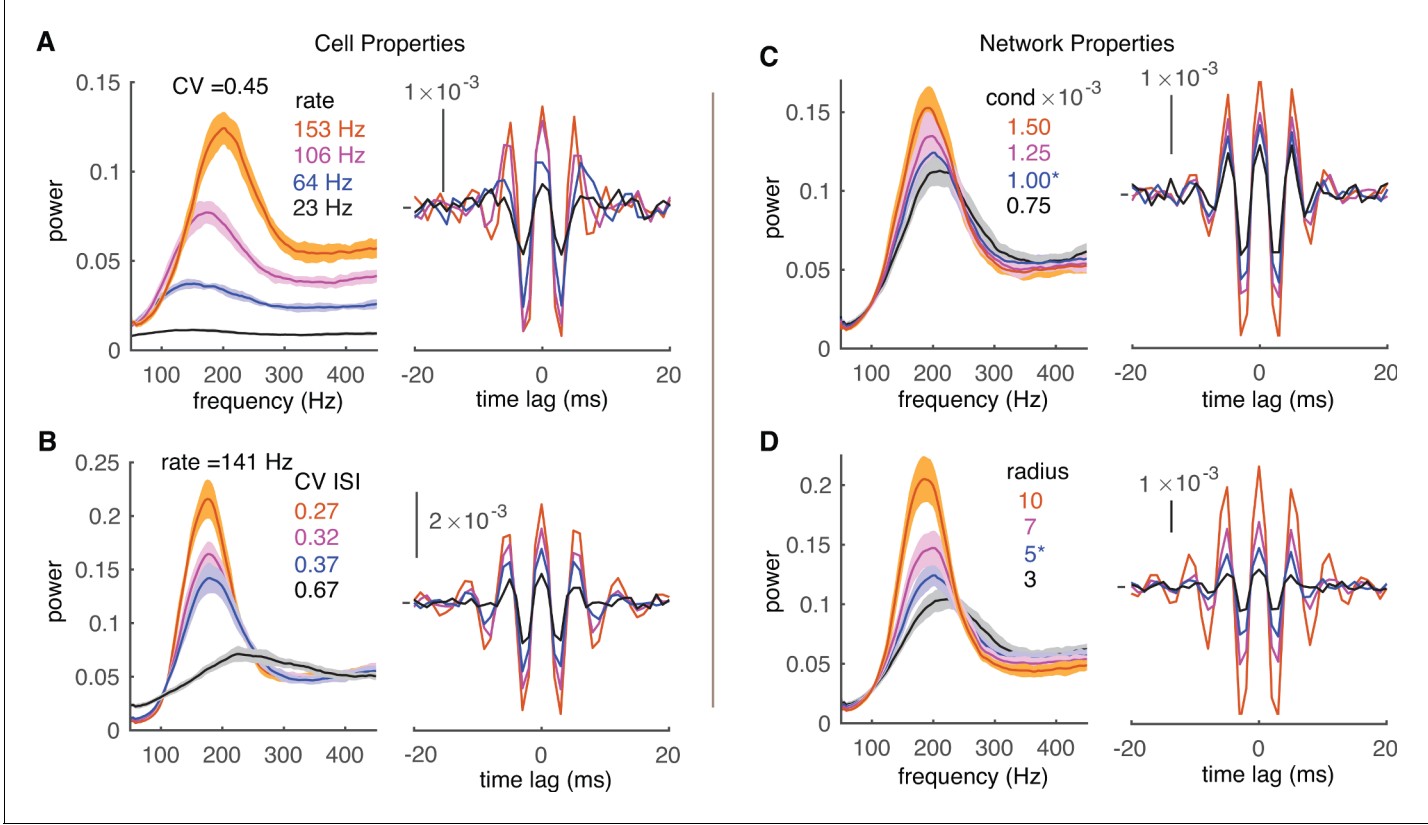

**Figure 4.** Effect of cell and network properties on high-frequency oscillations. (**A**) Low cellular firing rates decorrelate the network output in the forms of reduced peaks of power spectrums (left) and CCGs (right). CV ISI is ~0.45. Synaptic conductance is 1 nS and radius is 5. (**B**) Irregular spiking (high CV of ISIs) also decorrelates network. The cellular firing rate is ~141 Hz. Same layout and network properties as in A. (**C**) Small conductance (cond) of inhibitory synapses decorrelates network output. Same layout and network properties as in A with cellular firing rate ~151 Hz and CV ISI ~ 0.45. (**D**) Short connection radius also decorrelates network output. Same layout and cellular firing properties as C.

The online version of this article includes the following figure supplement(s) for figure 4:

**Figure supplement 1 .** Formation of High-frequency Oscillations at High Rates.

**Figure supplement 2 .** Decreased PRC at high firing rates can weaken oscillations.

firing rate specific PRC was used as the coupling term $Z(\theta)$ (see Materials and methods). In such a coupled oscillator network, the oscillation power shows a firing rate dependence similar to that of the complex PC network (**Figure 5A,B**). This finding demonstrates that the firing rate adaptation of the PRC is sufficient to cause firing-rate-dependent oscillations.

Next, we investigated the specific contribution of the flat part that dominates the PRC at low firing rates versus the late peak with increasing amplitude that appears at higher firing rates. We checked which of these PRC components is responsible for the effect on oscillations by fixing the amplitude of the peak to the value for a specific firing rate. Networks simulated with fixed PRC peak amplitudes show power spectra (**Figure 5C,D**) that are very similar to that obtained with the actual PRC (**Figure 5B**). An exception is when peak amplitude is very small (for firing frequencies of less than 30 Hz, not shown). The only significant difference between **Figure 5C and D** is the peak oscillation frequency, which increases with the firing rate for which the amplitude was taken.

In conclusion, the ratio of flat part width to peak width of the firing rate dependent PRC causes the rate dependence of high-frequency oscillations. At low firing rates the dominant flat part suppresses the coupling between oscillators. At high firing rates the coupling increases during the late peak and synchronizes the oscillators, but the strength of oscillation does not depend on peak amplitude in this network.

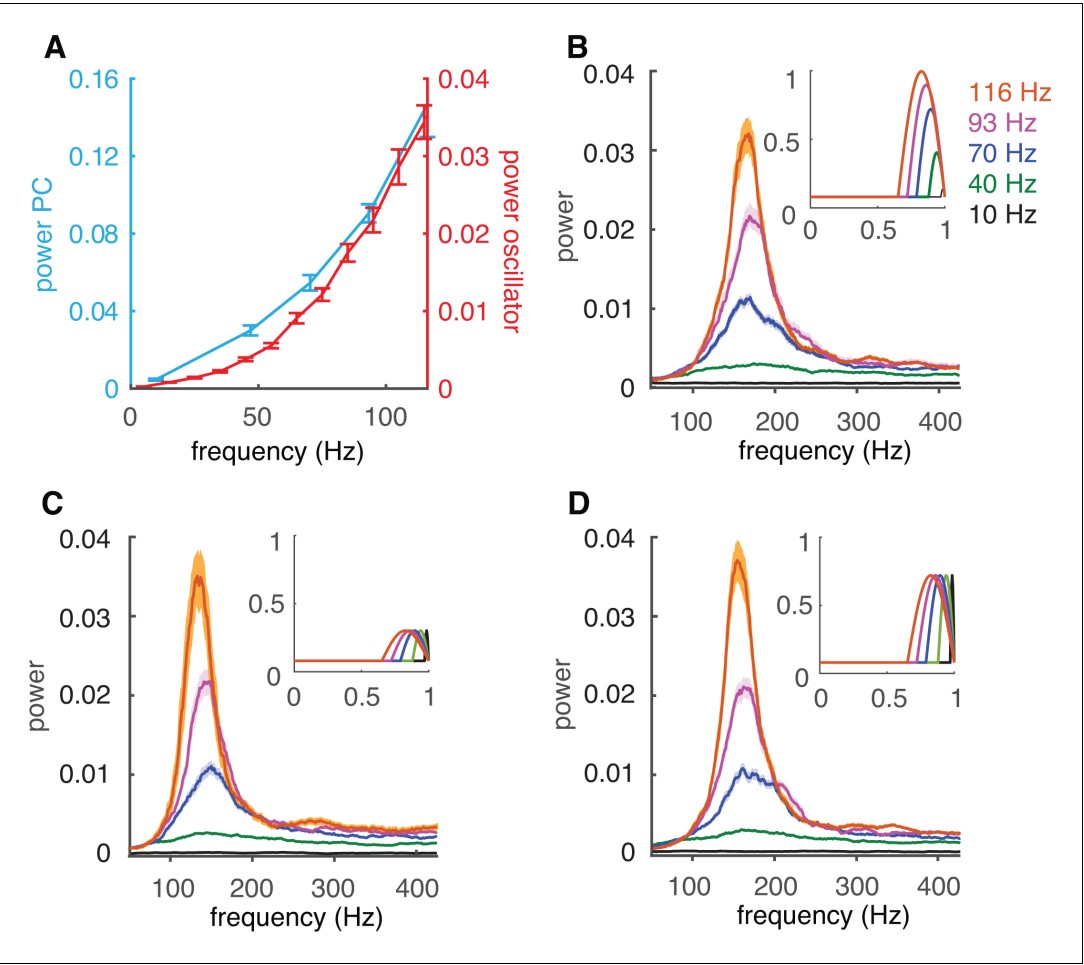

**Figure 5.** Firing-rate adaptation of high-frequency oscillations is caused by the PRC. (**A**) Dependence of peak power of high-frequency oscillations in the complex PC network of *Figure 3* (cyan) and in the coupled oscillator network (red) on cellular firing rate. (**B**) The power spectrum of the coupled oscillator network depends on the cellular firing-rate-specific PRC used as coupling term. Inset: firing-rate-dependent coupling factors $Z(\theta)$ used. (**C**) Same as B but with the peak amplitude of $Z(\theta)$ set to that of the peak of 30 Hz firing rate. (**D**) Same as C for the peak of 70 Hz firing rate.

## Transient correlations form cell assemblies

Correlation of spiking has often been proposed as a mechanism to form transient cell assemblies (*Abeles, 1982*; *Hebb, 1949*; *Singer, 1993*). This assumes that oscillations can appear and fade rapidly and that they can appear in networks with heterogeneous firing rates. We have previously simulated networks with a range of homogeneous stable cellular rates. Here, we first test whether rate-dependent synchrony still holds when population rates change dynamically. Population rates of the network approximate the half-positive cycle of a 1 Hz sine wave (peak ~140 Hz) with the duration of each trial being 1 s (*Figure 6A*). We compute shuffle-corrected, normalized joint peristimulus time histograms (JPSTHs) to reflect the dynamic synchrony (*Aertsen et al., 1989*; *Figure 6—figure supplement 1A*). The main and the third diagonals of the JPSTH matrix, corresponding to 0 ms-time lag correlation and 6 ms-time lag correlation respectively, are plotted to show the dynamic synchrony at transiently increased rates (bin size is 2 ms, *Figure 6B*). At low basal rates, there are no correlations between spikes. Both correlations start to increase ~250 ms after the onset of simulations when the cellular firing rate increases. Closely following rate changes, they decrease again when the cellular rates drop. It demonstrates that axon collateral-caused spike correlations can be achieved transiently to transmit a correlation code conjunctive with temporal cellular firing rate increases.

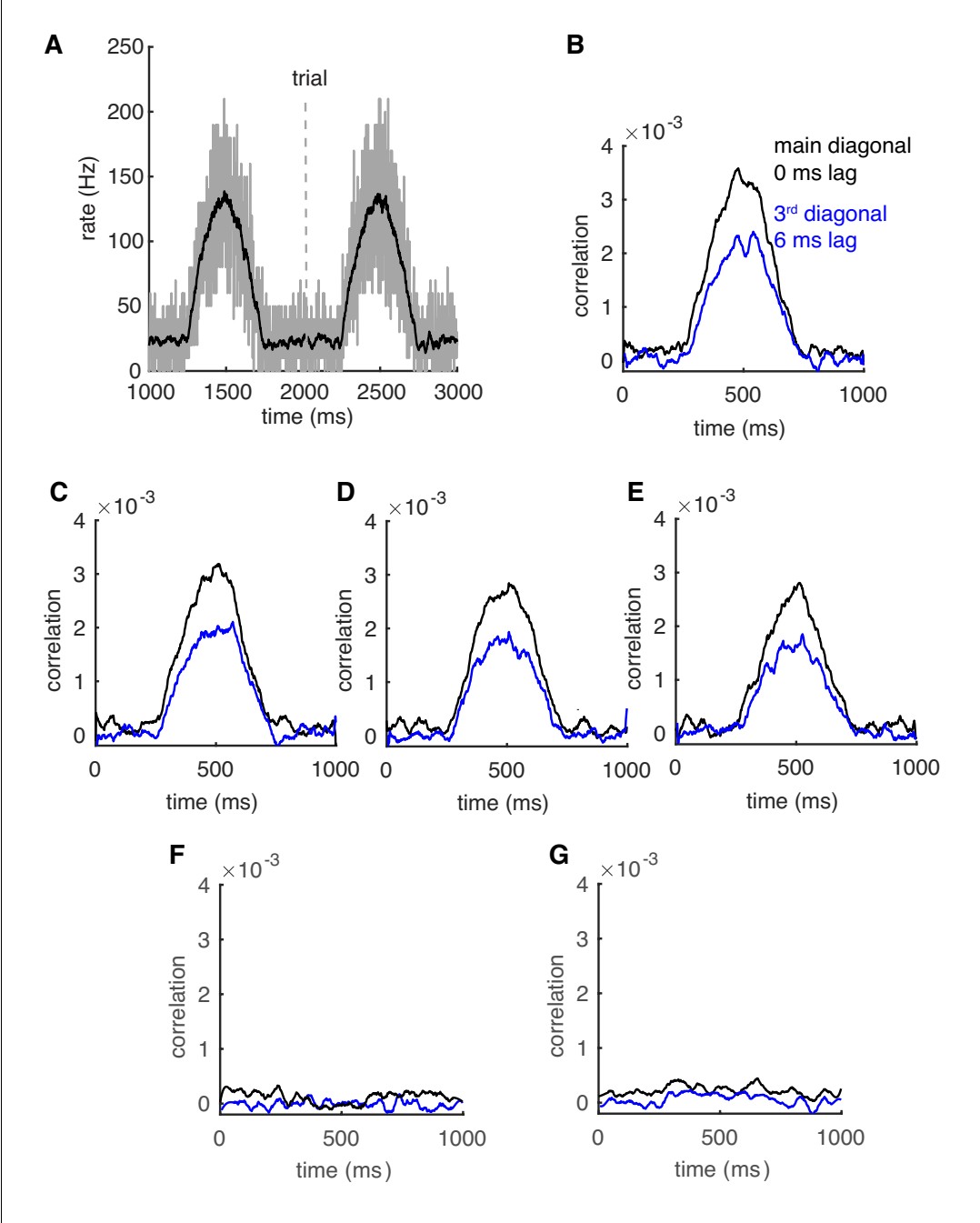

**Figure 6.** Correlations can be transient and robust to heterogeneous spike rates. (**A**) Population spike rates of PCs. (**B**) The 0 ms- and 6 ms-time lag correlations increase with population rates. (**C–E**) The rate-dependent correlation is robust to heterogeneous cellular rate changes. From (**C**) to (**E**), the number of decreased rate cells increases from 10 to 30. (**F**) Correlations between decreased-rate neurons in the network (n = 30). (**G**) Correlations between increased-rate neurons and decreased-rate neurons (n = 30 for each group).

The online version of this article includes the following figure supplement(s) for figure 6:

**Figure supplement 1 .** Dynamic correlations of the PC network outputs.

**Figure supplement 2 .** Transient correlations for a 2.5 Hz sine wave.

Although it remains unclear whether the population of PCs converging onto a same cerebellar nuclei (CN) neuron are homogeneous or heterogeneous (*Uusisaari and De Schutter, 2011*), simultaneous bidirectional PC rate changes have been observed during cerebellum-related behaviors (*Chen et al., 2016*; *Herzfeld et al., 2015*). It is very likely that neighboring PCs show heterogeneous

spike rate changes (*Hong et al., 2016*), which can reduce spike correlations (*Markowitz et al., 2008*). Therefore, we distributed 10–30 extra cells with decreasing spike rates (*Figure 6—figure supplement 1B*) in the network to test the effect of heterogeneous neighboring rate changes on transient correlations. They were randomly scattered among the cells with increasing rates. Spike correlations still become larger for the subgroup of PCs showing increased cellular rates, despite a slight decrease of the correlation amplitude when more cells decrease their spike rates (*Figure 6C–E*). Moreover, the spiking in PCs with decreased firing rates is not correlated (*Figure 6F*), nor is it correlated with oscillating increased-rate PCs (*Figure 6G*), making the assembly formation specific to fast spiking PCs. Similar results were obtained for a faster change of population rates (2.5 Hz sine wave, *Figure 6—figure supplement 2*). The results suggest that a population of PCs with increased spike rates can form a correlated assembly that will strongly affect downstream neurons even when it is surrounded by non-correlated neighboring PCs with decreased spike rates.

## Discussion

In this work, we reproduced the firing rate-dependent PRC of PCs and dissected the underlying bio-physical mechanisms. Next, we explored the role of these PRCs in synchronizing spikes in cerebellar PCs and how they can support the formation of transient assemblies.

### Biophysical mechanisms underlying rate-dependent PRCs

The profiles of neuronal PRCs are regulated by ionic currents (*Ermentrout et al., 2012*) and they show rate adaptation (*Couto et al., 2015*; *Ermentrout et al., 2001*; *Gutkin et al., 2005*; *Phoka et al., 2010*; *Tsubo et al., 2007*). Cerebellar PCs exhibit a transition from small, phase-independent responses to large, phase-dependent type-I responses with increasing rates (*Couto et al., 2015*; *Phoka et al., 2010*), but the mechanism was unknown (*Akemann and Knöpfel, 2006*; *Couto et al., 2015*; *De Schutter and Bower, 1994*; *Khaliq et al., 2003*; *Phoka et al., 2010*). This work reproduces and explains the experimentally observed rate adaptation of PRCs. Note that the slight increase of PRCs in the very narrow late phase in our model (low rate, *Figure 1B*) may be annihilated by noise in spontaneously firing neurons (*Couto et al., 2015*; *Phoka et al., 2010*).

Compared with previous work emphasizing the slow deactivation of $K^+$ currents in cortical neurons (*Ermentrout et al., 2001*; *Gutkin et al., 2005*), here we demonstrate the role of rate-dependent subthreshold membrane potentials and their corresponding activation of $Na^+$ channels. In both pyramidal neurons and PCs, spike rates cause significant variation of the subthreshold membrane potential during the ISI (*Rancz and Häusser, 2010*; *Tsubo et al., 2007*). In response to a stimulus, both $Na^+$ and $K^+$ currents are activated. In PCs, the main $K^+$ current is high-threshold activated (*Martina et al., 2003*; *Zang et al., 2018*); therefore, depolarization-facilitated $Na^+$ currents dominate, causing larger normalized PRCs at high rates (*Figure 2*). This facilitation may be further boosted in PCs by enhanced excitability, such as SK2 down-regulation reducing the AHP and elevating subthreshold membrane potentials (*Grasselli et al., 2020*; *Ohtsuki and Hansel, 2018*). We did not explore possible PRC differences between zebrin-positive and zebrin-negative PCs due to a lack of data (*Zhou et al., 2014*). Previous PC models (*Akemann and Knöpfel, 2006*; *Couto et al., 2015*; *De Schutter and Bower, 1994*; *Khaliq et al., 2003*; *Phoka et al., 2010*) included low-threshold-activated $K^+$ currents, which counteract facilitated $Na^+$ currents. In the original Traub model, slow deactivation of $K^+$ currents and consequent hyperpolarization synergistically reduce the normalized PRC peaks at high rates (*Ermentrout et al., 2001*; *Gutkin et al., 2005*). By minimally modifying the Traub model, elevated subthreshold membrane potentials generate larger normalized PRC peaks at high rates (*Figure 2—figure supplement 1*).

### The evidence supporting rate-dependent correlations

Rate-dependent synchrony in the cerebellum has been demonstrated for Golgi cells (*van Welie et al., 2016*) but not, as yet, for PCs. However, careful analysis of previous experimental data in the cerebellum provides some evidence to support our findings. In the work of *de Solages al., 2008*, units with lower average rates (<10 Hz) did not exhibit significant correlations between neighboring PCs, for unknown reasons. This can be explained by the small flat PRCs at low rates. Under extreme conditions, when the PRC is constantly 0 (equivalent to disconnection), no correlations can be achieved (*Figures 3–6*). Additionally, the experimental oscillation power increased by the application

of WIN 55,212–2, which was intended to suppress background excitatory and inhibitory synapses (*de Solages et al., 2008*). The increased power could be due to more regular spiking after inhibiting the activity of background synapses (*Figure 4B*). However, it could also be caused by increased spike rates (*Figure 4A*), because this agent also blocks P/Q type $Ca^{2+}$ channels and consequently P/Q type $Ca^{2+}$-activated $K^+$ currents, to increase spike rates (*Fisyunov et al., 2006*). Similarly, enhanced oscillations have also been observed in calcium-binding protein gene KO mice, which have significantly higher simple spike rates (*Cheron et al., 2004*). A more systematic experimental study of the firing rate dependent appearance of loose simple spike synchrony among PCs and its relation to behavior would be required to confirm these predictions.

The rate-dependent correlations observed in this study are different from those reported previously by *de la Rocha et al., 2007*. In that study, common input-mediated correlation increased rapidly with increasing rate at low firing rates in pyramidal cells (their Figure 1e) and in integrate-and-fire models (their Figure 2c), while the PRC mediated correlations in our study of inhibitory coupling only appear at much higher firing rates (*Figure 5A*). Moreover, the findings of *de la Rocha et al., 2007* are not general, they only apply for neurons with integrator firing properties (*Hong et al., 2012*).

## Down-stream effects of PC assemblies

PCs inhibit their target neurons in the CN, which in turn form the only cerebellar output. It is difficult to finely regulate CN firing rates with inhibition only, because it operates over the narrow voltage range between resting potentials and $GABA_A$ reversal potentials. Two solutions for this problem have been proposed. The first is that synchronized pauses of PC firing will release CN neurons from inhibition, leading to rebound firing (*De Schutter and Steuber, 2009*; *Lee et al., 2015*). There is strong evidence that this mechanism works in controlling the onset of movement in the conditioned eyeblink reflex (*Heiney et al., 2014*) and in saccade initiation (*Hong et al., 2016*). The other solution provides a more continuous rate modulated CN output by time-locking of CN spikes to PC input. Several experimental studies have demonstrated that partial synchronization of afferent PC spiking can time-lock the spikes of CN neurons to their input (*Gauck and Jaeger, 2000*; *Person and Raman, 2012*). The ability to rapidly increase the correlation level within a subgroup of PCs with increased firing rates (*Figure 6*) is therefore predicted to have a strong effect on CN spiking. Moreover, this does not require strong synchronization. Similar results were observed when jitter higher than the few ms predicted by our network model (*Figure 3B*) was applied to the synchronous PC input (*Gauck and Jaeger, 2000*). Previous evidence has demonstrated neocortical oscillations can entrain cerebellar oscillations (*Ros et al., 2009*). Though high-frequency oscillations (*Figure 3*) don't rely on common input, they can still be regulated by cortical inputs and drive neurons in the thalamocortical circuit (*Timofeev and Steriade, 1997*) and cerebral cortex (*Popa et al., 2013*).

## Advantages of transient PC assemblies

The actual convergence and divergence of PC axons onto CN neurons remains a controversial topic in the literature. There are roughly ten times more PCs than CN neurons and PC axons branch extensively leading to computed convergence values ranging from 20 to over 800 (*Uusisaari and De Schutter, 2011*), although many authors have recently converged on the compromise of ~50 (*Person and Raman, 2012*). If CN neurons just average the activity of all afferent PCs, much of the potential information generated by the large neural expansion in cerebellar cortex would be lost. Our PC network with parameters that fall within physiological ranges can rapidly generate and disrupt oscillations based on the cellular firing rates (*Figures 3* and *6*), with no need of increasing afferent input correlation. Note that rate-related synchrony can also be achieved via common synaptic inputs (*Heck et al., 2007*), gap junctions (*Middleton et al., 2008*), and ephaptic coupling (*Han et al., 2018*), when connections are weak. This means that transiently correlated PC assemblies can form and disappear quickly. Such assemblies, even if consisting of only a few PCs (*Person and Raman, 2012*), can finely control spiking in CNs. Because the assemblies can consist of variable subsets of afferent PCs to a CN neuron, this greatly expands the information processing capacity of the cerebellum.

## Conclusion

We have shown that firing-rate dependent PRCs can cause firing-rate dependent oscillations at the network level. Such a mechanism supports the rapid formation of transient neural assemblies in cerebellar cortex.

# Materials and methods

The detailed PC model and the interconnected network model were implemented in NEURON 7.5 (*Carnevale and Hines, 2006*). The Traub model was implemented in MATLAB.

## PRC computations

Our recently developed compartment-based PC model was used (*Zang et al., 2018*). To compute the PRCs in *Figure 1*, brief current pulses with a duration of 0.5 ms and an amplitude of 50 pA were administered at the soma at different phases of interspike intervals. The resulting perturbed periods were then used to calculate phase advances by *Ermentrout et al., 2001*:

$$PRC = (<ISI> - ISI_{perturb}) / <ISI> \qquad (1)$$

This is the same equation used in experimental studies (*Couto et al., 2015*; *Phoka et al., 2010*), to facilitate comparison. Different cellular rates were achieved by somatic holding currents (*Couto et al., 2015*; *Phoka et al., 2010*). To compute PRCs in response to negative stimuli, the amplitudes of the pulses were changed to -50 pA. To compute PRCs of our PC model with passive dendrites, only H current and leak current were distributed on the dendrites with the same parameters as in the active model (*Zang et al., 2018*). The Traub model (*Traub et al., 1999*) was implemented according to the work of *Ermentrout et al., 2001*; *Gutkin et al., 2005*. In the modified version of this model, the conductance of the kdr current was reduced from 80 to 40. Activation and deactivation rates of this current were shifted to the right by 30 mV, $\alpha_n(v)=0.032*(v+22)/(1-exp(-(v+22)/5))$; $\beta_n(v)=0.5*exp(-(v+27)/40)$; the conductance of AHP current was increased from 0 to 0.1.

## Network simulations

We implemented our recurrent inhibitory PC layer network using the Watts-Strogatz model (*Watts and Strogatz, 1998*) to avoid boundary effects. To reduce simulation time, we used the PC model with passive dendrites, which exhibits similar rate-dependent PRCs to the PC model with active dendrites (*Figure 1—figure supplement 1B*). In the baseline version of the network, 100 PCs were distributed on the parasagittal plane (*Witter et al., 2016*), corresponding to 2 mm of folium with a distance of 20 μm between neighboring PC soma centers. 100 PCs are within the estimated range of PCs converging to a same cerebellar nuclei neuron (*Person and Raman, 2012*). Each PC was connected to its nearest 2*radius neighboring PC somas and connections had 0 rewiring probability. The PCs were interconnected, according to anatomical data showing collaterals present toward both the apex and the base of the lobule with only slight directional biases (*Witter et al., 2016*). The baseline value of radius was 5 within the range of experimental estimates (*Bishop and O'Donoghue, 1986*; *de Solages et al., 2008*; *Watt et al., 2009*; *Witter et al., 2016*). The inhibitory postsynaptic current (IPSC) was implemented using the NEURON built-in point process, Exp2Syn. G = weight * (exp(-t/τ$_2$) - exp(-t/τ$_1$)), with τ$_1$ = 0.5 ms (rise time) and τ$_2$ = 3 ms (decay time). The reversal potential of the IPSC was set at −85 mV (*Watt et al., 2009*). The conductance was 1 nS (*de Solages et al., 2008*; *Orduz and Llano, 2007*; *Witter et al., 2016*). The delay between onset of an IPSC and its presynaptic spike timing was 1.5 ms (*de Solages et al., 2008*; *Orduz and Llano, 2007*; *Witter et al., 2016*). To test the effect of rate-dependent PRCs on high-frequency oscillations, we varied the cellular rates in two paradigms. In the first paradigm (*Figure 3*), each PC is contacted by 4000 excitatory parallel fiber synapses (PF, on spiny dendrites), 18 inhibitory stellate cells (STs, on spiny dendrites) and four inhibitory basket cells (BSs, on the soma). Activation of excitatory and inhibitory synapses in each PC was approximated as an independent Poisson process with different rates. We simulated five conditions: PC rate = 10 Hz when PF rate = 0.27 Hz, ST rate = 14.4 Hz, BS rate = 14.4 Hz; PC rate = 47 Hz when PF rate = 1.62 Hz, ST rate = 28.8 Hz, BS rate = 28.8 Hz (used in *Figure 5*); PC rate = 70 Hz when PF rate = 2.16 Hz, ST rate = 28.8 Hz, BS rate = 28.8 Hz; PC

rate = 93 Hz when PF rate = 2.7 Hz, ST rate = 28.8 Hz, BS rate = 28.8 Hz (used in *Figure 5*); PC rate = 116 Hz when PF rate = 3.24 Hz, ST rate = 28.8 Hz, BS rate = 28.8 Hz.

To more systematically explore different factors regulating network outputs, we used a second paradigm (*Figure 4*, *Figure 4—figure supplements 1* and *2*). Cellular rates of each PC were manipulated by injecting stochastic currents on the soma. The stochastic current was approximated by the commonly used Ornstein-Uhlenbeck random process (*Destexhe et al., 2001*), $\tau \frac{dI}{dt} = -I + \sigma\sqrt{\tau}\,\eta_i(t)$. $\sigma$ represents the amplitude of the fluctuation; $\eta_i$ represents uncorrelated white noise with unit variance; $\tau = 5\ \mathrm{ms}$. In this paradigm, we systematically varied the rates and firing irregularities of PCs (CV of ISIs) to explore their importance for network output. Due to the intrinsic relationship between CV of ISIs and firing rates, a larger $\sigma$ is required for higher firing rates to get the same CV of ISI. Phase response is a result of input current and response gain of the cell. We reduce the phase response by halving the input current (synaptic conductance) to achieve a smaller response at high firing rates (*Figure 4—figure supplements 2*). The conductance of inhibitory synapses was tested with the values of 0.75, 1.0, 1.25 and 1.5 nS in *Figure 4C*. We also explored the effect of connection radius with the values of 3, 5, 7 and 10 in *Figure 4D*.

To test a spatio-temporally increased correlation, we randomly distributed extra 10–30 PCs with decreased cellular rates into the original network (*Figure 6*, *Figure 6—figure supplement 2*), including 100 increased-rate cells. Similar with *Figure 4*, each PC receives dynamic current injections approximated by an Ornstein-Uhlenbeck random process. Their mean population firing rates are shown in *Figure 6—figure supplement 1B*.

## Coupled oscillator model

The model comprises 100 neurons that are randomly connected to each other with connection probability of p=0.75 (*Figure 5*). The 'subthreshold dynamics' of individual neurons is given by the phase equation

$$\frac{d\theta}{dt} = \frac{1}{T} + Z(\theta)(s_{net}(t) + s_{ind}(t)),$$

where $\theta$ is a phase variable ranging from 0 to 1. $T$ is an intrinsic period of the oscillation. $Z(\theta)$ is a PRC. $s_{ind}$ and $s_{net}$ are the individual and network input, respectively. At $\theta=1$, the model cell "spiked." Then, $\theta$ was reset to $\theta-1$ and the spike was added to the spike train variable (see below).

$Z(\theta)$ is given by

$$Z(\theta) = \begin{cases} A \cdot c & \text{if } 0 \le \theta \le 1\text{-}\delta, \\ A(c + B\sin(\pi(\theta + \delta - 1)/\delta)) & \text{if } 1\text{-}\delta < \theta \le 1. \end{cases}$$

Here, $c$ represents the flat part of the Purkinje cell PRC and the other term represents a 'bump' around $\theta = 1$. We found that the bump width is ~3 ms in time regardless of the firing rate, and set $\delta = 3\ \text{ms}/T$. We also used $c = 0.08$ and $A = 12$.

In the case when the model PRC scales as the PC PRC (*Figure 5B*), $B = f_{amp}(1/T)$ where $f_{amp}(r)$ is a normalized PRC amplitude given a baseline firing rate $r$ in *Figure 1C*. In *Figure 4C and D* with no amplitude scaling of the PRC, $B = f_{amp}(30\ \text{Hz})$ and $B = f_{amp}(70\ \text{Hz})$ are used, regardless of $T$, respectively.

$s_{ind}(t)$ is given by the Ornstein-Uhlenbeck (OU) process, $\partial_t s_{ind} = -s_{ind}/\tau + \sigma_0\zeta$, where $\zeta$ is a Wiener process based on the standard normal distribution. We used $\tau = 3$ ms and $\sigma_0 = 0.2$.

$s_{net}(t)$ is given by

$$\frac{ds_{net}}{dt} = -\frac{s_{net}}{\tau_{syn}} + i(t),$$

$$i(t) = g\sum_j o_j(t - d),$$

where $j$ represents other neurons connected to each cell, and $o_j(t)$ is a spike train of the cell $j$. $d = 1.5$ ms is a synaptic delay. $g = -20$ is a connection weight, and $\tau_{syn} = 3$ ms is a decay time for the synaptic current.

We used the forward Euler method with a time step of 0.025 ms to integrate the subthreshold equation, while we also confirmed that our results did not change if we use 0.0125 ms. The OU processes were integrated with the same time step and backward Euler method.

## Data analysis

The power spectrum of the spike trains of the network was estimated by Welch's method, which calculates the average of the spectra of windowed segments (window size 128 points). In each trial under each specific stimulus condition, the length of the signal was 2 s, with a time resolution of 1 ms. The final result was the average of 14 trials.

To compute the CCGs under each specific stimulus condition, we first computed pairwise correlations between the spike trains of two neurons and then corrected them by shift predictors, which removed the 'chance correlations' due to rate changes. Then correlations were divided by the triangular function $\Theta(\tau) = T - |\tau|$ and $\sqrt{\lambda_i \lambda_j}$. T was the duration of each trial and $\tau$ was the time lag. $\Theta(\tau)$ corrects for the degree of overlap between two spike trains for each time lag $\tau$. $\lambda_i$ was the mean firing rate of neuron $i$ (*Kohn and Smith, 2005*). Finally, the CCGs between all pairs in the network were averaged to reflect the population level spike correlations. Thus, similar with previous work (*Heck et al., 2007*), the computed CCGs reflect the 'excess' correlation caused by axon collaterals in our work.

To measure the dynamic correlation over the time course of the stimulus, we computed JPSTHs (*Aertsen et al., 1989*). We first picked two neurons from our network and aligned their spike-count PSTHs to stimulation onset with 2 ms time bins in each trial (larger time bins annihilated the positive peaks due to the significant negative correlations in paired spikes, see CCGs in *Figures 3* and *4*). We constructed the JPSTH matrix by taking each stimulus trial segment and plotting the spike counts of one cell on the horizontal and one on the vertical. If there is a spike from neuron i at time *x*, and a spike from neuron j at time *y*, one count will be added to the matrix index (x,y). By repeating this process for different trials, we got a raw matrix for a cell pair i and j. Then by the shift-predictor (repeated previous steps with shuffled stimulation trials), we removed correlations due to co-stimulation caused firing rate changes. Next step, we normalized the JPSTH by dividing with the product of standard deviations of the PSTHs of each neuron. To measure the correlation of the assembly, we averaged JPSTH between all non-repeated cell pairs in the defined 'assembly' of our network (*Oemisch et al., 2015*). The corrected matrix values become correlation coefficients, with values between −1 and +1. The main diagonal of the JPSTH matrix provides a measure of time-varying 0 ms time lag correlations and the third main diagonal (2 ms time bin) provides a measure of 6 ms time lag correlations. Due to the small-time bin we used, we simulated 1992 trials (for *Figure 6B–E*) to compute JPSTH between PC pairs and smoothed the JPSTHs for visualization purpose. Due to the small number of decreased-rate neurons in the network, we simulated 26112 trials to compute *Figure 6F,G* (30 decreased-rate neurons). When decreased-rate neuron numbers are 10 and 20 (*Figure 6C,D*), we did not compute their correlations due to the computational challenge. For *Figure 6G*, we randomly picked 30 from 100 increased-rate neurons to make pairs with 30 decreased-rate neurons.

## Acknowledgements

The authors thank for helpful suggestions from Drs. Eve Marder, Tomoki Fukai and Sergio Verduzco to improve the manuscript and for the language editing by Steven Douglas Aird.

## Additional information

### Funding

| Funder | Author |
| --- | --- |
| Okinawa Institute of Science and Technology Graduate University | Erik De Schutter |

The funders had no role in study design, data collection and interpretation, or the decision to submit the work for publication.

## Author contributions

Yunliang Zang, Conceptualization, Data curation, Software, Formal analysis, Validation, Investigation, Visualization, Methodology, Writing - original draft, Writing - review and editing; Sungho Hong, Software, Investigation, Visualization, Methodology, Writing - review and editing; Erik De Schutter, Supervision, Writing - original draft, Project administration, Writing - review and editing

## Author ORCIDs

Yunliang Zang (iD) https://orcid.org/0000-0001-8999-1936
Sungho Hong (iD) https://orcid.org/0000-0002-6905-7932
Erik De Schutter (iD) https://orcid.org/0000-0001-8618-5138

## Decision letter and Author response

Decision letter https://doi.org/10.7554/eLife.60692.sa1
Author response https://doi.org/10.7554/eLife.60692.sa2

# Additional files

## Supplementary files

- Source code 1. Purkinje_PRC_Oscillation_Code.
- Transparent reporting form

## Data availability

All computer codes used to generate simulation data in this work have been provided.

The following dataset was generated:

| Author(s) | Year | Dataset title | Dataset URL | Database and Identifier |
|---|---|---|---|---|
| Zang Y, Hong S, De Schutter E | 2020 | Firing rate-dependent phase responses of Purkinje cells support transient oscillations | http://modeldb.yale.edu/266799 | ModelDB, 266799 |

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
