## [Decision Letter]

**Acceptance summary:**

This modeling study addresses an important problem in cellular and network dynamics. First, it provides a biophysical mechanism underlying the changes in the shape of the phase response curve (PRC) of Purkinje cells observed when they change their (simple spike) firing rate. Second, it demonstrates that this mechanism depends on the subthreshold voltage trajectory between spikes, which in turn is governed by the intrinsic biophysics of the neurons. Finally, the authors show that firing rate-dependent changes in the PRC of individual neurons can drive rate-dependent changes in network synchrony. This new mechanism is likely to regulate synchrony of high-frequency oscillations not just in the cerebellum but also in many other circuits in the brain.

The study is carefully designed and performed, and the manuscript is well written. Overall it makes an important contribution to how specific features of single-cell biophysics can help to determine network-level dynamics.

**Decision letter after peer review:**

[Editors’ note: the authors resubmitted a revised version of the paper for consideration. What follows is the authors’ response to the first round of review.]

Thank you for submitting your work entitled "Firing rate-dependent phase responses of purkinje cells support transient oscillations" for consideration by *eLife*. Your article has been reviewed by a Senior Editor, a Reviewing Editor, and three reviewers. The following individuals involved in review of your submission have agreed to reveal their identity: Bard Ermentrout (Reviewer #1).

Our decision has been reached after consultation between the reviewers. Based on these discussions and the individual reviews below, we regret to inform you that your work will not be considered further for publication in *eLife*.

There was a strong agreement on the importance of PRCs for understanding neuronal activity and for the ambitions of the work presented. Several substantive concerns about the execution of the work arose, and there was a concern that the relationship of the PCR to network function was not established limiting the impact of the findings. At the end of this decision letter you will see the de-identified consultation among reviewers for this manuscript. This consultation in conjunction with the written reviews could serve as the basis for a complete revision of the current manuscript that could serve as a new submission to *eLife*.

Reviewer #1:

I was a bit surprised at this result until I realized what the authors had done with respect to the computation of the PRC. In the infinitesimal limit,the PRC is the normalized nullspace of the adjoint linear operator. The usual normalization is that the inner product is 1. This is how the theory that the authors allude to is done. Here the authors use a different normalization where they divide the adjoint by the period. This produces a different scaling and of course will make the PRC that occurs near the bifurcation (where the period is large) to appear much smaller in amplitude than it would with the standard scaling. If you use the standard scaling on the Traub model (which is readily available) and use the authors modifications, you find that with the standard normalization, higher frequencies have smaller PRCs which is according to the theory. If you divide by the period, then you get the authors results at least for currents that are not too big. For frequencies of about 110 Hz, even with the scaling, I found the PRC to be smaller than the lower frequencies.

Turning to their PC model, it is clear from Figure 1C, that the amplitudes with the usual normalization will follow the theory. That is, higher-frequencies have smaller amplitudes. (Specifically, if you divide the amplitudes shown by their respective frequencies, the graph will be decreasing.)

The main motive for this study is the PRC measurements from Couto et al. which was published in PLoS comp bio where the same result was found experimentally. To me the amplitude is not so much the point of interest here but rather the extreme change in the shape of the PRC. The Traub model does not show the shape dependence in fact it is the opposite with little effects until the late phases for low frequencies, an effect that was explained in the paper by Pascal et al. as coming from the adaptation.

The main result of interest is that at higher rates, oscillations synchronize/correlate better than an low rates So, there is some practical effect of this.

The authors set out to explain several effects. One of them is the largely unresponsive phase after the spike. As noted above, this effect was already explained in the old work of Pascal et al.

In Figure 3D, while the amplitude goes up in this model, the phase dependence is very flat only in the low rate versions. Thus, I think that the authors explanations are not really general. They need to make these explanations in a more rigorous mathematical manner.

The latter part of the paper is the most interesting where they demonstrate rate dependence in the correlation of the networks. It is a shame that they do not directly relate this to the PRCs. It would have been nice to perform some sort of analysis of a network of neurons that viewed as phase oscillators and connected via their phase response curves. I should point out that when coupling, whether or not locking occurs depends on whether or not you rescale the heterogeneties or not. I illustrate this in the following example. Let I be the baseline current to a bunch of neural oscillatora and let I_j be the heterogeneous current (assumed small) Let T be the period and suppose simple pulse coupling such that is neuron j fires, then a delta function current J_{ij} is added to the postsynaptic current. Let Delta be the conventional PRC (normalized to be msec/mv). Then the Kuramoto/Ermentrout/Kopell/ etc reduction yields for the phase:

dtheta_i/dt = 1 + Delta (theta_i)[ I_i/C + sum J_{ij}/C delta(t-t_j)] mod T

with C the membrane capacitance. Theta is a time-like variable at this point. To make it a phase-like variable (as the authors do), then let theta = T phi. Then one obtains

dphi_i/dt = omega + (1/T) Delta(T phi_i) [ I_i/C + sum J_{ij}/C delta(t-t_j)] mod 1

Conditions for synchrony etc are the same as the first model if, say you reduce this to a phase-difference model by replacing the delta(t) with delta(phi).

My point is, that the amplitude dependence on frequency is partially a consequence of how you normalize the PRC. If you look at Tsubo Figure 4, the amplitudes are all the same in their normalized version. The theory predicted by the theta model is that the period-normalized PRCs should have constant amplitudes.

I think the results are interesting, but I also think that they need to do some more sophisticated math analysis and also relate their PRCs more directly to the synachronization/correlation effects. This could be done with a weak coupling analysis after convolving their PRCs with the synapses to obtain the interaction function. The most important facet of PRCs is not som much their amplitude (which can be compensated for by changing the coupling), but their shape. I'd like more theoretical understanding of the shape dependence.

Reviewer #2:

This manuscript explores how average firing rate in Purkinje cells affects their PRC and high-frequency oscillations. Overall, there are issues with presentation, especially for clarity and the significance of the findings.

Essential revisions:

1) The manuscript, as presented, is disjointed. The first half proposes a mechanistic factor that contributes to rate modulation in Purkinje cell PRCs, and has a clear conclusion (although not many other mechanisms are entertained). The second half describes how the average firing rate increases the power of high frequency oscillations in the cerebellum. But there is not a strong link between these halves. In particular, it is not clear if it is the same mechanism underlying both types of rate adaptation.

2) It is not clear why the results of the study are significant, especially considering the relatively broad readership of *eLife*. The authors did not link rate adaptation to Purkinje cell or cerebellar function, and the claim that their results hold "for any type of neuron" (subsection “General Effect of Subthreshold Membrane Potentials on Shaping PRCs”) doesn't quite work because only two particular models of two types of neuron were studied.

3) In the second half of the manuscript, the relationship between the form of loose synchrony studied here, and correlations is unclear. This is important because it is well established that correlations increase with firing rate (de la Rocha, 2007, not cited in the manuscript). Does this explain what the authors are observing? (Particularly in the last figure).

4) Figure legends are too brief and do not include the relevant information to understand the figure.

5) The writing, especially of the Introduction and Discussion section, are overly specialized. I do not think that a reader unfamiliar with the PRC would find this manuscript accessible.

Reviewer #3:

This modeling study addresses an important problem in cellular and network dynamics. First, it provides a biophysical mechanism underlying the changes in the shape of the phase response curve (PRC) of Purkinje cells observed when they change their (simple spike) firing rate. Second, it demonstrates that this mechanism depends on the subthreshold voltage trajectory between spikes, which in turn is governed by the intrinsic biophysics of the neurons. Finally, the authors show that firing rate-dependent changes in the PRC of individual neurons can drive rate-dependent changes in network synchrony. This new mechanism is likely to regulate synchrony of high-frequency oscillations not just in the cerebellum but also in many other circuits in the brain.

The study is carefully designed and performed, and the manuscript is well written. Overall, it makes an important contribution to how specific features of single-cell biophysics can help to determine network-level dynamics.

Essential revisions:

1) The authors mention briefly in subsection “PRC Exhibits Rate Adaptation in PCs” that "Rate-adaptive PRCs require the presence of a dendrite in the PC model (not shown), but the dendrite can be passive (Figure 1—figure supplement 1B)", without going into further detail. In their explanation of the biophysical mechanism of rate adaptation of PRCs in PCs that follows (in subsection “PRC Exhibits Rate Adaptation in PCs”), dendrites are not mentioned, however, as if the dendrite was not relevant to the mechanism at all. Did the authors follow this somewhat contradictory strategy because they consider the role of the dendrite too simple or too complicated to explain? Could the main consequence of the presence of an active or passive dendrite be its (capacitive and Ohmic) load, leading to a depolarizing shift of the somatic voltage threshold of spikes (Bekkers and Hausser, 2007; Zang et al., 2018)? With the consequence that, in a well-tuned, physiologically detailed PC model like that of the authors, removal of the dendrite would lead to a shift in the spike threshold in the hyperpolarized direction, artificially interfering with the mechanism illustrated by the authors in Figure 2A and B?

If the importance of the dendrite is in fact due to an effect of this (or a similar) kind, then not only would the reader benefit from a brief explanation, but the authors could also make experimentally testable predictions of what happens to the PRC at different firing rates when the (passive or active) dendrite is pinched, i.e. isolating the some from the dendrite.

2) More generally, can the authors propose other experimentally testable predictions resulting from their biophysical mechanism of rate-dependent PRCs? This would help strengthen the study.

3) I am somewhat baffled by the words "just the passive depolarization" in the explanation (subsection “The Biophysical Mechanism of Rate Adaptation of PRCs in PCs”) that "During early phases of all rates, membrane potentials are distant from the Na^+^ activation threshold (Figure 2A,B). The depolarizations to weak stimuli fail to activate sufficient Na^+^ channels to speed up voltage trajectories, and phase advances are caused by just the passive depolarizations (Figure 2C). Consequently, phase advances in early phases are small and flat (or phase independent)." In the bottom (12 Hz) PRC in Figure 1B, there is an (admittedly broad) local maximum near phase 0.2. If these phase advances due to stimuli in an early phase of the PRC were indeed caused just by the resulting passive depolarizations, then the amplitude of these depolarizations should decay with the passive membrane time constant, leading to smaller PRC amplitudes at early phases (such as 0.2) than later phases (such as 0.8). The 12 and 27 Hz PRCs in Figure 1B show the opposite effect, suggesting that the membrane potential of a passive soma is not the only relevant state variable governing the approximately flat part of the PRC. Which other state variables (e.g. dendritic membrane potential, calcium concentration, activation or inactivation state of ion channels) could explain the shape of the 'foot' of the PRC at low rates?

[Editors’ note: further revisions were suggested prior to acceptance, as described below.]

Thank you for submitting your article "Firing Rate-dependent Phase Responses of Purkinje Cells Support Transient Oscillations" for consideration by *eLife*. Your article has been reviewed by 3 peer reviewers, and the evaluation has been overseen by Ronald Calabrese as the Senior Editor, a Reviewing Editor, and three reviewers. The following individuals involved in review of your submission have agreed to reveal their identity: Bard Ermentrout (Reviewer #2).

The reviewers have discussed the reviews with one another and the Reviewing Editor has drafted this decision to help you prepare a revised submission.

Summary:

This modeling study addresses an important problem in cellular and network dynamics. First, it provides a biophysical mechanism underlying the changes in the shape of the phase response curve (PRC) of Purkinje cells observed when they change their (simple spike) firing rate. Second, it demonstrates that this mechanism depends on the subthreshold voltage trajectory between spikes, which in turn is governed by the intrinsic biophysics of the neurons. Finally, the authors show that firing rate-dependent changes in the PRC of individual neurons can drive rate-dependent changes in network synchrony. This new mechanism is likely to regulate synchrony of high-frequency oscillations not just in the cerebellum but also in many other circuits in the brain.

Essential revisions:

Reviewer #3:

The reviews by the three other referees have already appropriately summarized the findings and commented on the modeling aspects of the study. For this reason, I would like to restrict myself to a brief discussion of cell physiological aspects of the work. Overall, the study is well done, and I believe that this work will be important to the field of cerebellar physiology, with further reaching implications in the neurosciences regarding the impact of neuronal oscillations.

1) It should be stated in the results section whether the modeling focuses on Purkinje cells in adult animals, or during development. This is crucial information, keeping in mind that the nature of Purkinje cell – Purkinje cell interactions changes during development (see Watt et al., 2009; cited).

2) Results section: the cell's responsiveness and spike output (in response to synaptic drive) appear to change with the state of the AHP, not only the amplitude of synaptic input (Ohtsuki et al., 2018). Does the model predict how the oscillatory phase affects synaptically driven spike firing?

3) Are resurgent Na conductances (Raman and Bean, 1997) critical for the occurrence of these oscillations or specific parameters?

4) Does the model account for differences in spike firing frequencies in zebrin-positive and zebrin-negative cerebellar modules (Zhou et al., 2014)? This is suggested by the findings of Schonewille and others (Grasselli et al., 2020) that in SK channel knockout mice these firing properties are differentially affected, which highlights the role of the AHP in sub-and suprathreshold modulation.

5) Are oscillations in the cerebellum coherent with high-frequency oscillations in other brain areas? This is suggested by the observation that input from the cerebellar nuclei regulates gamma frequency oscillations in thalamocortical networks (Timofeev and Steriade, 1997).

Addressing these points will shed more light on the consequences of these oscillations for cerebellar output functions and will thus hopefully further enhance the impact of this interesting paper.

---

## [Author Response]

[Editors’ note: the authors resubmitted a revised version of the paper for consideration. What follows is the authors’ response to the first round of review.]

Reviewer #1:I was a bit surprised at this result until I realized what the authors had done with respect to the computation of the PRC. In the infinitesimal limit,the PRC is the normalized nullspace of the adjoint linear operator. The usual normalization is that the inner product is 1. This is how the theory that the authors allude to is done. Here the authors use a different normalization where they divide the adjoint by the period. This produces a different scaling and of course will make the PRC that occurs near the bifurcation (where the period is large) to appear much smaller in amplitude than it would with the standard scaling. If you use the standard scaling on the Traub model (which is readily available) and use the authors modifications, you find that with the standard normalization, higher frequencies have smaller PRCs which is according to the theory. If you divide by the period, then you get the authors results at least for currents that are not too big. For frequencies of about 110 Hz, even with the scaling, I found the PRC to be smaller than the lower frequencies.

We use a standard normalization for PRCs computed for experimental data or, in our case, complex model data where it is close to impossible to compute the adjoint (for more than 1000 coupled ODEs). Equation 1 we used to compute the PRC is identical to the equation in Ermentrout, Pascal and Gutkin (2001). As far as we know, the usual normalization the reviewer alludes to can only be applied to very simple models. We now mention explicitly that the use of Equation 1 facilitates comparison with experimental data (subsection “PRC Computations”).

We do understand the broader argument that this standard normalization of the PRC, dividing the difference in spike timing by the period, promotes a firing rate-dependence of the peak amplitude of the PRC. This has never been explicitly acknowledged in previous experimental literature that described this phenomenon, namely Phoka et al., 2010 and the confirmation by Couto et al., 2015. Therefore, we mention this issue now explicitly in subsection “PRC Exhibits Rate Adaptation in PCs” of the new manuscript.

It should be noted that, by itself, this normalization does not always increase the peak amplitude of the PRC as demonstrated in Figures 1—figure supplement 2 and Figure 1—figure supplement 1A where, using Equation 1, peak amplitude decreases with firing rate.

Turning to their PC model, it is clear from Figure 1C, that the amplitudes with the usual normalization will follow the theory. That is, higher-frequencies have smaller amplitudes. (Specifically, if you divide the amplitudes shown by their respective frequencies, the graph will be decreasing.)The main motive for this study is the PRC measurements from Couto et al. which was published in PLoS comp bio where the same result was found experimentally. To me the amplitude is not so much the point of interest here but rather the extreme change in the shape of the PRC.

This is a very astute remark. We show indeed later in the new manuscript that the important PRC property is largely the phase width of the flat part relative to the phase width of the peak (new Figure 5 and subsection “High-frequency Oscillations are Caused by the Rate-dependent PRC”).

The Traub model does not show the shape dependence in fact it is the opposite with little effects until the late phases for low frequencies, an effect that was explained in the paper by Pascal et al. as coming from the adaptation.The main result of interest is that at higher rates, oscillations synchronize/correlate better than an low rates So, there is some practical effect of this.The authors set out to explain several effects. One of them is the largely unresponsive phase after the spike. As noted above, this effect was already explained in the old work of Pascal et al.In Figure 3D, while the amplitude goes up in this model, the phase dependence is very flat only in the low rate versions. Thus, I think that the authors explanations are not really general. They need to make these explanations in a more rigorous mathematical manner.

We thank the reviewer for agreeing with the importance of firing rate-dependent PRCs on the better oscillations at higher rates by ‘there is some practical effect of this’. Traub model is less emphasized in the new manuscript and has completely been moved to supplementary material. We corrected statements about the model according to these comments.

The latter part of the paper is the most interesting where they demonstrate rate dependence in the correlation of the networks. It is ashame that they do not directly relate this to the PRCs. It would have been nice to perform some sort of analysis of a network of neurons that viewed as phase oscillators and connected via their phase response curves. I should point out that when coupling, whether or not locking occurs depends on whether or not you rescale the heterogeneties or not. I illustrate this in the following example. Let I be the baseline current to a bunch of neural oscillatora and let I_j be the heterogeneous current (assumed small) Let T be the period and suppose simple pulse coupling such that is neuron j fires, then a δ function current J_{ij} is added to the postsynaptic current. Let Δ be the conventional PRC (normalized to be msec/mv). Then the Kuramoto/Ermentrout/Kopell/ etc reduction yields for the phase:dtheta_i/dt = 1 + Δ(theta_i)[ I_i/C + sum J_{ij}/C δ(t-t_j)] mod Twith C the membrane capacitance. Theta is a time-like variable at this point. To make it a phase-like variable (as the authors do), then let theta = T phi. Then one obtainsdphi_i/dt = omega + (1/T) Δ(T phi_i) [ I_i/C + sum J_{ij}/C δ(t-t_j)] mod 1Conditions for synchrony etc are the same as the first model if, say you reduce this to a phase-difference model by replacing the δ(t) with δ(phi).

We thank the reviewer for agreeing this part ‘is the most interesting’ and for this excellent suggestion. We implement a coupled oscillator model and show the results in new Figure 5 and accompanying new subsection “High-frequency Oscillations are Caused by the Rate-dependent PRC”. Thanks to the coupled oscillator model, we can now formally show that the rate-dependent PRC directly causes the rate-dependent oscillations and that the flat part of the PRC plays an important role in this property.

My point is, that the amplitude dependence on frequency is partially a consequence of how you normalize the PRC. If you look at Tsubo Figure 4, the amplitudes are all the same in their normalized version. The theory predicted by the theta model is that the period-normalized PRCs should have constant amplitudes.

Explicitly mentioned in the new manuscript in subsection “PRC Exhibits Rate Adaptation in PCs”.

I think the results are interesting, but I also think that they need to do some more sophisticated math analysis and also relate their PRCs more directly to the synachronization/correlation effects. This could be done with a weak coupling analysis after convolving their PRCs with the synapses to obtain the interaction function. The most important facet of PRCs is not som much their amplitude (which can be compensated for by changing the coupling), but their shape. I'd like more theoretical understanding of the shape dependence.

See comments above about new Figure 5 and the importance of PRC shape.

Reviewer #2:This manuscript explores how average firing rate in Purkinje cells affects their PRC and high-frequency oscillations. Overall, there are issues with presentation, especially for clarity and the significance of the findings.Essential revisions:1) The manuscript, as presented, is disjointed. The first half proposes a mechanistic factor that contributes to rate modulation in Purkinje cell PRCs, and has a clear conclusion (although not many other mechanisms are entertained). The second half describes how the average firing rate increases the power of high frequency oscillations in the cerebellum. But there is not a strong link between these halves. In particular, it is not clear if it is the same mechanism underlying both types of rate adaptation.

We agree with this criticism, as argued in subsection “High-frequency Oscillations are Caused by Rate-dependent PRCs” this was the best we could do with the complex PC network model we used. Based on a suggestion of reviewer #1 we have now also implemented a coupled oscillator model that allows us to demonstrate a causal link between the rate-dependent PRC and the rate-dependent oscillations. This is described in a new section “High-frequency Oscillations are Caused by the Rate-dependent PRC” (subsection “High-frequency Oscillations are Caused by Rate-dependent PRCs” and new Figure 5). The manuscript has also been reorganized to focus mostly on the cerebellum and to emphasize this causal link (we moved the work on the Traub model to supplementary material).

2) It is not clear why the results of the study are significant, especially considering the relatively broad readership of eLife. The authors did not link rate adaptation to Purkinje cell or cerebellar function, and the claim that their results hold "for any type of neuron" (subsection “General Effect of Subthreshold Membrane Potentials on Shaping PRCs”oesn't quite work because only two particular models of two types of neuron were studied.

Reviewer #3 does believe the paper ‘addresses an important problem in cellular and network dynamics’. But we agree that the original manuscript did a rather poor job of explaining, based on the work of Person and Raman, (2012), why transient synchrony can be important in the cerebellum. We now explicitly refer to the formation of transient assemblies, discuss how such correlated assemblies can change the firing of downstream neurons and why their transient nature expands the information processing capacity of the cerebellum (Discussion section). We also expanded the analysis of the 1 Hz wave data of Figure 6 and generated new data (2.5 Hz wave, Figure 2—figure supplement 2) to support the hypothesis of transient assemblies.

3). In the second half of the manuscript, the relationship between the form of loose synchrony studied here, and correlations is unclear. This is important because it is well established that correlations increase with firing rate (de la Rocha, 2007, not cited in the manuscript). Does this explain what the authors are observing? (Particularly in the last figure).

Based both on mechanism (common excitatory input for de la Rocha model, inhibitory coupling reinforced by firing-rate dependent PRC in our model) and actual shape of the firing rate dependence (‘square root’ like in de la Rocha model, ‘exponential’ like for our model, now shown in new Figure 5A), we believe that our work describes a fundamentally different phenomenon. This is now discussed in subsection “The Evidence Supporting Rate-dependent Correlations”.

4) Figure legends are too brief and do not include the relevant information to understand the figure.

We improved several legends.

5) The writing, especially of the Introduction and Discussion section, are overly specialized. I do not think that a reader unfamiliar with the PRC would find this manuscript accessible.

Introduction and Discussion section were largely rewritten.

Reviewer #3:This modeling study addresses an important problem in cellular and network dynamics. First, it provides a biophysical mechanism underlying the changes in the shape of the phase response curve (PRC) of Purkinje cells observed when they change their (simple spike) firing rate. Second, it demonstrates that this mechanism depends on the subthreshold voltage trajectory between spikes, which in turn is governed by the intrinsic biophysics of the neurons. Finally, the authors show that firing rate-dependent changes in the PRC of individual neurons can drive rate-dependent changes in network synchrony. This new mechanism is likely to regulate synchrony of high-frequency oscillations not just in the cerebellum but also in many other circuits in the brain.The study is carefully designed and performed, and the manuscript is well written. Overall, it makes an important contribution to how specific features of single-cell biophysics can help to determine network-level dynamics.

We thank the reviewers for acknowledging the importance of our work.

Essential revisions:1). The authors mention briefly in subsection “PRC Exhibits Rate Adaptation in PCs” that "Rate-adaptive PRCs require the presence of a dendrite in the PC model (not shown), but the dendrite can be passive (Figure 1—figure supplement 1B)", without going into further detail. In their explanation of the biophysical mechanism of rate adaptation of PRCs in PCs that follows (in subsection “PRC Exhibits Rate Adaptation in PCs”), dendrites are not mentioned, however, as if the dendrite was not relevant to the mechanism at all. Did the authors follow this somewhat contradictory strategy because they consider the role of the dendrite too simple or too complicated to explain? Could the main consequence of the presence of an active or passive dendrite be its (capacitive and Ohmic) load, leading to a depolarizing shift of the somatic voltage threshold of spikes (Bekkers and Hausser, 2007; Zang et al., 2018)? With the consequence that, in a well-tuned, physiologically detailed PC model like that of the authors, removal of the dendrite would lead to a shift in the spike threshold in the hyperpolarized direction, artificially interfering with the mechanism illustrated by the authors in Figure 2A and B?If the importance of the dendrite is in fact due to an effect of this (or a similar) kind, then not only would the reader benefit from a brief explanation, but the authors could also make experimentally testable predictions of what happens to the PRC at different firing rates when the (passive or active) dendrite is pinched, i.e. isolating the some from the dendrite.

As already shown in Zang et al., 2018 the dendrite extensively changes the spike shape, without dendrite there is a stronger afterhyperpolarization that significantly changes the subthreshold trajectory. As a consequence, the PRC without dendrite looks completely different. See new Figure 1—figure supplement 2 and subsection “Biophysical Mechanisms Underlying Rate-dependent PRCs”.

2) More generally, can the authors propose other experimentally testable predictions resulting from their biophysical mechanism of rate-dependent PRCs? This would help strengthen the study.

We now added suggestions to the Discussion section.

3) I am somewhat baffled by the words "just the passive depolarization" in the explanation (subsection “The Biophysical Mechanism of Rate Adaptation of PRCs in PCs”) that "During early phases of all rates, membrane potentials are distant from the Na^+^ activation threshold (Figure 2A,B). The depolarizations to weak stimuli fail to activate sufficient Na^+^ channels to speed up voltage trajectories, and phase advances are caused by just the passive depolarizations (Figure 2C). Consequently, phase advances in early phases are small and flat (or phase independent)." In the bottom (12 Hz) PRC in Figure 1B, there is an (admittedly broad) local maximum near phase 0.2. If these phase advances due to stimuli in an early phase of the PRC were indeed caused just by the resulting passive depolarizations, then the amplitude of these depolarizations should decay with the passive membrane time constant, leading to smaller PRC amplitudes at early phases (such as 0.2) than later phases (such as 0.8). The 12 and 27 Hz PRCs in Figure 1B show the opposite effect, suggesting that the membrane potential of a passive soma is not the only relevant state variable governing the approximately flat part of the PRC. Which other state variables (e.g. dendritic membrane potential, calcium concentration, activation or inactivation state of ion channels) could explain the shape of the 'foot' of the PRC at low rates?

We no longer use “passive depolarization” as it may be a confusing terminology. We emphasize the importance of activation of transient versus persistent sodium channels (subsection “Subsection “The Biophysical Mechanism of Rate Adaptation of PRCs in PCs”).

[Editors’ note: what follows is the authors’ response to the second round of review.]

Essential revisions:Reviewer #3:The reviews by the three other referees have already appropriately summarized the findings and commented on the modeling aspects of the study. For this reason, I would like to restrict myself to a brief discussion of cell physiological aspects of the work. Overall, the study is well done, and I believe that this work will be important to the field of cerebellar physiology, with further reaching implications in the neurosciences regarding the impact of neuronal oscillations.

We thank the reviewer for agreeing with the importance of this work.

1) It should be stated in the Results section whether the modeling focuses on Purkinje cells in adult animals, or during development. This is crucial information, keeping in mind that the nature of Purkinje cell – Purkinje cell interactions changes during development (see Watt et al., 2009; cited).

We added text to the Results section to clarify that we simulated adult cerebellar cortex and the evidence to support the presence of collateral connections in adults.

2) Results section: the cell's responsiveness and spike output (in response to synaptic drive) appear to change with the state of the AHP, not only the amplitude of synaptic input (Ohtsuki et al., 2018). Does the model predict how the oscillatory phase affects synaptically driven spike firing?

Previous theoretical work cited in the Introduction established that the PRC predicts how the oscillatory phase affects spike firing caused by synaptic input. As the model reproduces the experimental PRC, it should also make this prediction. The paper has been cited and discussed in Subsection “Biophysical Mechanisms Underlying Rate-dependent PRCs”.

3) Are resurgent Na conductances (Raman and Bean, 1997) critical for the occurrence of these oscillations or specific parameters?

Yes, resurgent Na conductance is critical for PRCs and then for oscillations. We used a state model of sodium channel gating from Raman and Bean, 2001, which incorporates transient, resurgent and persistent current in one model. We now mentioned resurgent component in subsection “The Biophysical Mechanism of Rate Adaptation of PRCs in PCs”.

4) Does the model account for differences in spike firing frequencies in zebrin-positive and zebrin-negative cerebellar modules (Zhou et al., 2014)? This is suggested by the findings of Schonewille and others (Grasselli et al., 2020) that in SK channel knockout mice these firing properties are differentially affected, which highlights the role of the AHP in sub-and suprathreshold modulation.

Due to lack of data, we can’t account for differences in zebrin-positive and zebrin-negative cerebellar modules. We have clearly stated this as a limitation. These papers have been cited and discussed. Subsection “Biophysical Mechanisms Underlying Rate-dependent PRCs”.

5) Are oscillations in the cerebellum coherent with high-frequency oscillations in other brain areas? This is suggested by the observation that input from the cerebellar nuclei regulates γ frequency oscillations in thalamocortical networks (Timofeev and Steriade, 1997).

We discuss coherence of oscillations in subsection “Down-stream effects of PC assemblies”.